# Proteotoxicity caused by perturbed protein complexes underlies hybrid incompatibility in yeast

Krishna B. S. Swamy [1,3,4], Hsin-Yi Lee[1,4], Carmina Ladra[1], Chien-Fu Jeff Liu[1], Jung-Chi Chao[1], Yi-Yun Chen [2] & Jun-Yi Leu [1] ✉

Dobzhansky–Muller incompatibilities represent a major driver of reproductive isolation between species. They are caused when interacting components encoded by alleles from different species cannot function properly when mixed. At incipient stages of speciation, complex incompatibilities involving multiple genetic loci with weak effects are frequently observed, but the underlying mechanisms remain elusive. Here we show perturbed proteostasis leading to compromised mitosis and meiosis in *Saccharomyces cerevisiae* hybrid lines carrying one or two chromosomes from *Saccharomyces bayanus* var. *uvarum*. Levels of proteotoxicity are correlated with the number of protein complexes on replaced chromosomes. Proteomic approaches reveal that multi-protein complexes with subunits encoded by replaced chromosomes tend to be unstable. Furthermore, hybrid defects can be alleviated or aggravated, respectively, by up- or down-regulating the ubiquitin-proteasomal degradation machinery, suggesting that destabilized complex subunits overburden the proteostasis machinery and compromise hybrid fitness. Our findings reveal the general role of impaired protein complex assembly in complex incompatibilities.

Proteins are the main functional units and building blocks of the cell. To survive in constantly changing environments, cells maintain protein homeostasis (proteostasis) by a complex network of translation, molecular chaperone, and degradation systems[1,2]. Perturbed proteostasis can have strong impacts on cell physiology and has been implicated in aging and many different human diseases[3,4]. Proteins usually execute their activities through a multitude of interactions with other proteins, as in the case of the protein complexes[5] that control most biological processes or functions[6,7]. In the budding yeast, *Saccharomyces cerevisiae*, about half of the proteome are subunits of protein complexes[6,8].

Individual subunits within a complex often share similar evolutionary patterns, and alterations in partner proteins can cause decreased complex stability or even failures in complex assembly[9,10].

Since a substantial proportion of eukaryotic proteins (~30% in humans) can adopt folded or functional three-dimensional conformations only upon binding to their partner proteins[11], unassembled complex subunits need to be carefully monitored by the protein homeostasis machinery to prevent massive misfolded protein aggregations[12,13]. Thus, maintaining balanced protein complex dynamics represents a major challenge for the proteostasis system, especially in stress conditions[2].

Dobzhansky–Muller incompatibilities refer to deleterious genetic interactions between functionally diverged loci during the evolution of new species. They are widely accepted as one of the major mechanisms underlying hybrid sterility or inviability, i.e., the features associated with postzygotic reproductive isolation between different species[14,15]. Genetic incompatibilities have been discovered in yeast, animals, and

[1]Institute of Molecular Biology, Academia Sinica, Taipei 11529, Taiwan. [2]Institute of Biological Chemistry, Academia Sinica, Taipei 11529, Taiwan. [3]Present address: Division of Biological and Life Sciences, School of Arts and Sciences, Ahmedabad University, Ahmedabad 380009, India. [4]These authors contributed equally: Krishna B. S. Swamy, Hsin-Yi Lee. ✉e-mail: jleu@imb.sinica.edu.tw

plants[16,17], with most incompatibilities identified to date comprising two interacting components. Even though complex incompatibilities involving three or more loci are regularly found in experimental crosses[18–20], their functions at the molecular level have yet to be characterized. Such complex incompatibilities can yield insights into the processes of divergence, especially at the early stage of speciation[21]. Some complex incompatibilities have weak effects individually, but can synergistically cause hybrid breakdown with other incompatible loci[22]. Understanding complex incompatibility could thus reveal fundamental rules underlying the patterns and rate of reproductive isolation between diverging species.

Formation of a functional protein complex often involves interactions between different coevolved subunits and disturbed complex formation can easily lead to compromised physiology. Moreover, when individual incompatible subunits may only have a mild impact on the complex formation, collectively their effects can be synergistic. It raises an interesting possibility that if subunits from different parents have trouble forming functional complexes in hybrid cells, they may form a basis for complex incompatibility[23].

*S. cerevisiae* and *S. bayanus* var. *uvarum* (previously known as *S. bayanus*) are two closely related yeast species that diverged from a common ancestor about 20 million years ago[24]. There is no apparent prezygotic isolation between them under laboratory conditions, and hybrid strains have been isolated from wineries and natural environments[25,26]. However, strong postzygotic isolation exists between these two species. The spore viability of F1 hybrids is only 0.5% and any viable F2 progeny continuously suffers from compromised mitosis and meiosis[27,28]. We constructed 11 chromosome replacement lines in which one or two *S. cerevisiae* chromosomes are replaced by homologous chromosomes from *S. bayanus* var. *uvarum*[28]. These lines emulate the F1 gametes or F2 progeny in the wild and hence can help in understanding the natural mechanisms driving early speciation. Using them, we previously identified a strongly incompatible gene pair contributing to hybrid sterility[28], demonstrating that chromosome replacement lines are a useful system for dissecting genetic incompatibility.

Here, we use eleven different diploid chromosome replacement lines to understand the general consequences of mixed genomes in hybrid cells. We find that most replacement lines reveal a transcriptional signature of proteotoxic stress under normal growth conditions. The intrinsic proteotoxic stress compromises the ability of hybrid cells to adapt to changing external environments, resulting in reduced growth and sporulation rates. Moreover, levels of proteotoxic stress are positively correlated with the number of unique protein complexes encoded on replaced chromosomes. We further examine the proteomes of two most defective lines to demonstrate that proteotoxic stress is indeed caused by destabilized protein complexes, arising from improper or failed interactions between complex subunits encoded by the two different genomes. Finally, we show that by up- or downregulating the ubiquitin-proteasomal system we can alleviate or aggravate, respectively, the defects of replacement lines, providing evidence that perturbed proteostasis causes compromised fitness.

## Results

### Chromosome replacement lines exhibit a transcriptional signature of stress responses

Previously, we constructed 11 chromosome replacement lines between two closely related species, *S. cerevisiae* (Sc) and *S. bayanus* var. *uvarum* (Sb), to dissect the contribution of individual chromosomes to hybrid incompatibility[28]. Although we identified several cases of strong mitochondrial-nuclear incompatibilities involving two genes[28,29], the weak and polygenic nuclear-nuclear incompatibilities widely observed in these lines remained unexplored. Transcriptome analysis can represent a sensitive way of detecting hybrid dysfunctions[30–32]. Therefore, we evaluated the transcriptomic consequences of the presence of foreign chromosomes in all the 11 diploid replacement lines derived from crossing a and α types of the respective replacement lines. We conducted RNA sequencing on total RNA isolated from diploid cell lines grown at 23 °C, a non-stressful temperature for both *S. cerevisiae* and *S. bayanus* var. *uvarum*, and classified a gene as being differentially expressed if the fold-change in expression between the replacement line and the parental line (Sc) was greater than 1.5 with an adjusted $p$-value < 0.05.

We identified several hundred genes as differentially expressed in each replacement line, even when the genes on the replaced foreign chromosomes were excluded (Fig. 1a and Supplementary Data 1). Why would one or two foreign chromosomes from a closely related species have such a strong influence on the rest of the genome? We found that a large proportion of the differentially expressed genes were commonly up- or downregulated in multiple replacement lines, suggesting that these changes may represent a general response to foreign chromosomes (Fig. 1a and Supplementary Data 1). Moreover, many of these common-response genes encode molecular chaperones or proteins related to stress responses[33]. Transcriptome analyses of yeast cells grown under diverse exogenous stresses (such as heat shock, osmotic shock, starvation, and oxidative stress) revealed 868 genes that were commonly up- or downregulated under such stresses, termed ESR (environmental stress response) genes[33]. When we compared the expression profiles of ESR genes between our replacement lines and the cells subjected to various stresses, we observed a positive correlation between most of our replacement lines and stress-treated cells (Fig. 1b and Supplementary Data 2). That ESR signature suggests that our replacement lines harboring foreign chromosomes were physiologically stressed even though the cells were growing in a non-stressful environment. Such a transcriptomic stress response signature is not specific to our replacement lines and has been observed in hybrids of fungi, plants, and animals[34–39]. It suggests a general phenomenon in hybrids to cope up with physiological stress caused by the coexistence of two divergent genomes. Nonetheless, the detailed mechanisms remain elusive.

### Chromosome replacement lines display proteotoxic stress

What causes ESR induction in our chromosome replacement lines? Studies in aneuploid cells suggest that ESR can be induced intrinsically by the proteotoxic stress arising from unbalanced chromosome numbers[40,41]. Notably, the ESR signature in our replacement lines was significantly correlated to that observed for an aneuploid population ($\rho = 0.41$, $p < 2.23 \times 10^{-16}$, Spearman's rank correlation, Supplementary Data 2)[42]. However, the replacement lines are euploid and derived from related yeast species with an almost complete set of orthologous proteins. Thus, the cause of ESR in hybrid replacement lines is likely to be different from that in aneuploid strains.

Hsp104 is a protein disaggregase widely used as a marker for protein aggregation under many conditions of proteotoxic stress[43]. We assessed if our diploid replacement lines and normal F1 hybrid diploids containing a complete set of the two parental genomes also suffered from proteotoxic stress by analyzing the subcellular localization of Hsp104 during heat adaptation[41]. We cultured the yeast strains carrying the Hsp104-mCherry fusion protein at 23 °C and then shifted them to 37 °C to induce protein aggregation. We anticipated that, initially, the Hsp104-mCherry signal would be diffused throughout the cytosol, but that after heat treatment it would co-localize with protein aggregate foci to clear the aggregates[44]. Accordingly, cells suffering from intrinsic proteotoxic stress should have more pronounced protein aggregations and take a longer time to dissolve the aggregates before Hsp104 would disperse throughout the cytosol once again[41].

Indeed, our protein aggregation assay showed that all the tested replacement lines as well as the F1 hybrid diploids had a significantly higher proportion of cells with Hsp104 foci at an early time-point (45 min) and took longer to dissolve all aggregates compared to the *S.*

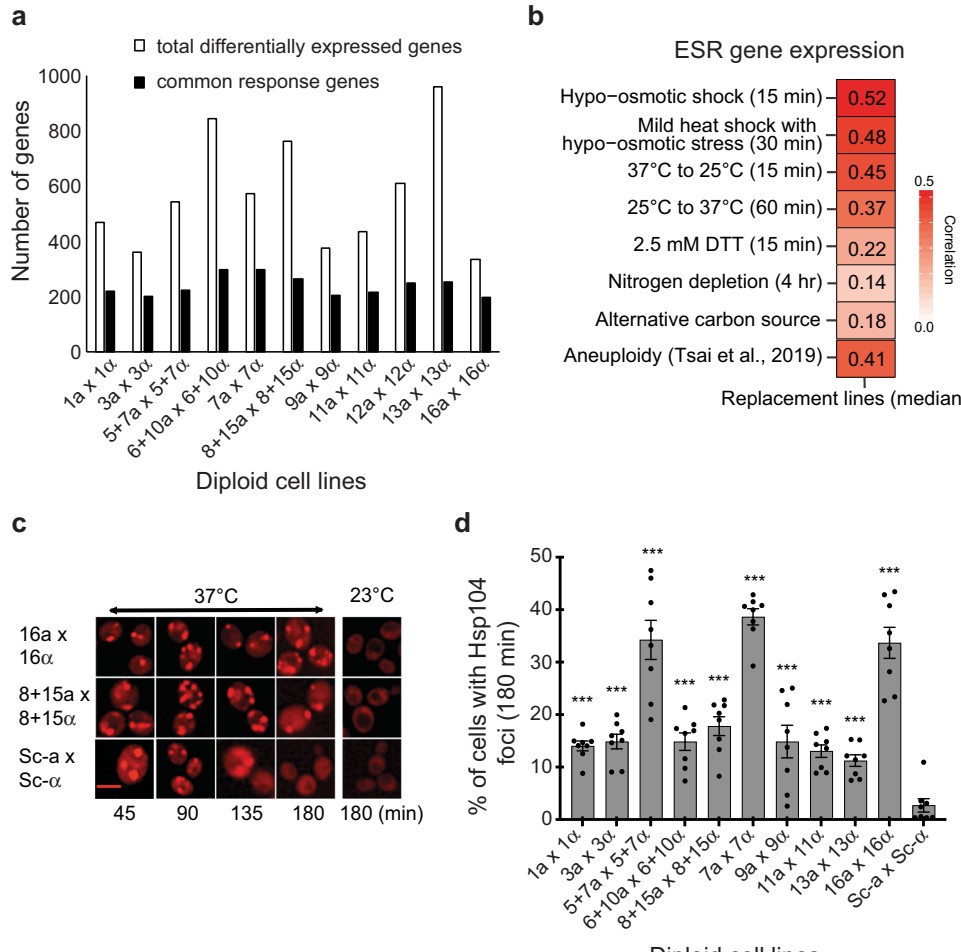

**Fig. 1 | Foreign chromosomes induce proteotoxic stress in diploid hybrid cells.**
**a** Hundreds of genes are differentially expressed in diploid chromosome replacement lines. The replacement lines were cultured in rich medium at 23 °C and their transcriptomes were examined using RNA-seq. A gene was classified as differentially expressed if the fold-change of a replacement line gene to its parental *S. cerevisiae* strain was greater than 1.5 with an adjusted *p*-value < 0.05. The genes on the replaced chromosomes were excluded from the analysis. Common-response genes refer to the genes commonly up- or downregulated in at least four replacement lines. A complete gene list can be found in Supplementary Data 1. **b** The expression profile of diploid replacement lines is positively correlated with those of wild-type cells under stress conditions and aneuploid cells. The median expression levels in eleven diploid replacement lines were compared to the environmental stress response (ESR) dataset[33] and the aneuploid transcriptome[42]. Spearman's correlation coefficients were calculated and are shown in the figure. Details of correlations for individual lines are presented in Supplementary Data 2. **c** Images of

Hsp104-mCherry aggregates in diploid *S. cerevisiae* (Sc) and replacement cell lines. Hsp104 was diffusely localized in the cytosol when cells were grown at 23 °C, but formed foci upon shifting the temperature to 37 °C. Scale bar: 5 μm. These results were reproducible in eight independent experiments. **d** Diploid replacement lines take longer to dissolve all aggregates upon heat treatments. Yeast strains containing an *HSP104-mCherry-URA* cassette were grown to exponential phase in YPD at 23 °C and then shifted to 37 °C. The percentage of cells containing Hsp104-mCherry foci was determined 180 min after shifting the temperature. The 12 L line was not included in this assay since *HSP104* is located on replaced Chromosome 12 and *S. bayanus* var *uvarum* was excluded as they are cryotolerant and cannot tolerate heat treatments ($n = 8$; $N \geq 500$ cells per time-point). Data are presented as mean values +/− SEM. ***: *p*-value < 10⁻³; one-sided Student's t-test between *S. cerevisiae* and replacement lines. Source data and detailed statistical information are provided as a Source Data file.

*cerevisiae* control (Fig. 1c and Supplementary Fig. 1a). At 180 min after the temperature shift, Hsp104 foci were undetectable in most *S. cerevisiae* control cells, but both the replacement lines and F1 hybrid diploids exhibited significant retention of aggregates (Fig. 1d and Supplementary Fig. 2). These results corroborate our transcriptome data that proteostasis is perturbed in cells carrying foreign chromosomes. Among our diploid replacement lines, *S. cerevisiae* cells carrying *S. bayanus* var. *uvarum* chromosomes 5 and 7 (5+7L), 8 and 15 (8+15L), and 16 (16L) exhibited the slowest adaptive kinetics, suggesting that these lines might display the most severe intrinsic proteotoxic stress. Since we had previously established that the 5+7L replacement line also suffers from mitochondrial-nuclear incompatibility[29], we focused our subsequent experimental analyses on the 8+15L and 16L lines.

To rule out the possibility that the defect in heat adaptation presented by hybrid cells was due to cell death or an inability to mount a heat shock response, we measured the cell viability of all lines and the protein abundance of several heat-induced molecular chaperones in the two selected replacement lines exhibiting the most severe phenotypes. Our results show that the replacement lines did not have obvious defects in viability or heat shock response at 37 °C (Supplementary Fig. 1b, c).

A recent study showed that the *S. cerevisiae* W303 strain was more sensitive to proteotoxicity than other natural isolates due to a defective *SSD1* allele in W303 and introducing *SSD1* from the oak soil strain YPS1009 into W303 restored proteotoxicity tolerance[45]. Since our replacement lines were derived from the W303 strain, we checked if the strain background was the primary source of the proteotoxicity by

introducing $SSD1^{YPS1009}$ into the most defective replacement lines. At 180 min after the temperature shift, $SSD1^{YPS1009}$-containing replacement lines still retained a significant percentage of cells harboring $HSP104$ foci, suggesting that the observed proteotoxic stress in hybrids was not simply due to the W303 strain background (Supplementary Fig. 3).

## Stress resulting from harboring foreign chromosomes causes mitotic and meiotic defects

Our protein aggregation assay revealed that the diploid replacement lines do not adapt to temperature changes as efficiently as the parental strain. However, only some replacement lines exhibited obvious growth defects under normal conditions (Fig. 2a)[28]. To further understand how the intrinsic proteotoxic stress induced by foreign chromosomes impacts cell fitness, we measured mitotic growth rates in rich medium containing a low dose of the Hsp90 inhibitor, geldanamycin (GdA). Hsp90 is a molecular chaperone essential for maintaining proteostasis and relieving the proteotoxicity caused by stress[46]. Moreover, several client proteins and protein complexes of Hsp90 are required for mitosis and meiosis[47]. Thus, Hsp90 inhibition is likely to enhance mild defects in mitosis and meiosis if they already exist in replacement lines. At 23 °C, mild interference of Hsp90 by GdA treatment (50 μM) did not cause obvious growth defects in *S. cerevisiae* and *S. bayanus* var. *uvarum* cells. However, we observed significantly reduced cell growth in all replacement lines as well as the F1 hybrid diploids (Fig. 2a and Supplementary Fig. 2c), suggesting that intrinsic proteotoxic stress further sensitized the cells to even slight perturbation of proteostasis. The growth defects were not specific to GdA or Hsp90 since we observed similar reductions in fitness when the replacement lines were grown at 32 °C, representing mild heat stress that did not affect the growth of wild-type cells (Fig. 2b and Supplementary Fig. 4a).

Hybrid sterility is often observed upon mating between two species. To examine the effect of intrinsic proteotoxic stress on meiosis, we perturbed the proteostasis of replacement lines with the same mild dosage of GdA before sporulation and then measured sporulation rates (see "Methods"). Two replacement lines (7L and 13L) were excluded from this assay since they already exhibited severe sporulation defects due to mitochondrial-nuclear incompatibility[28]. Among the nine tested replacement lines, six presented significantly higher sensitivity to the perturbation than the parental lines and exhibited significantly reduced sporulation (Fig. 2c). Consistent with our protein aggregation data, the 8+15L and 16L lines showed the most severe defects in both mitosis and meiosis among all tested replacement lines. These results demonstrate that although hybrid cells carrying foreign chromosome(s) only display weak or mild incompatibility, the fitness defects can be easily aggravated by mild environmental perturbations that are tolerable to wild-type cells.

## Levels of proteotoxicity are correlated with the number of protein complexes on replaced chromosomes

In our replacement lines, many orthologous proteins expressed from the foreign chromosomes substitute the functions of the respective endogenous proteins. One possible cause for the proteotoxic stress we observed is that the proteostasis-related proteins in *S. bayanus* var. *uvarum* are less efficient and more sensitive to environmental perturbations (described as the "weak allele" hypothesis). Alternatively, the proteotoxicity may be induced by misfolded protein complex subunits that have dissociated from unstable chimeric complexes or that failed to assemble[48]. In a previous study that examined the formation of six stable protein complexes in hybrids between *Saccharomyces* yeasts (*sensu stricto*), three of the six formed only species-specific complexes[49], indicating that species-specific interactions between complex subunits can evolve quickly, even among closely related species. Since the complex subunits encoded

by the replaced chromosome may require subunits encoded on other chromosomes to form protein complexes, any level of species-specific interactions would likely reduce the assembly efficiency or stability of chimeric complexes, resulting in an excess of unassembled or misfolded subunits (described as the "unstable complex" hypothesis).

To test these two hypotheses, we crossed 8+15L and 16L haploid cells with *S. bayanus* var. *uvarum* to generate heterozygous diploid cells and examined their fitness. A complete set of the *S. bayanus* var. *uvarum* chromosomes were present in these hybrid diploids, so genes on the replaced chromosomes were homozygous. These hybrid strains should still be sensitive to mild Hsp90 perturbations if the replaced chromosomes carry a "weak allele" of proteostasis-related genes. In contrast, if proteotoxicity is caused by unstable protein complexes, the fitness defect should be partially alleviated since the relative abundance of the unstable complexes is reduced. We observed significant rescue of fitness defects when Hsp90 was compromised (50 μM GdA, 23 °C), suggesting that the fitness defect of replacement lines is not due to a "weak allele" from the *S. bayanus* var. *uvarum* chromosomes (Fig. 2d). This is further supported in F1 hybrid diploids, where the fitness defect in the presence of one copy of interaction partners was significantly higher than the parental diploids, but reduced considerably when compared to 8+15L and 16L (Supplementary Fig. 2c).

Individual complex subunits often interact with more than one partner in protein complexes. Therefore, compromised interactions between different subunits can result in complex epistatic effects, as observed in complex incompatibility[7]. Moreover, the loading effect leading to proteotoxicity is cumulative even though the contribution of each unassembled (or misfolded) complex subunit may be mild. We tested if the level of proteotoxic stress in each replacement line is correlated with the number of unique protein complexes encoded by the replaced chromosome (see "Methods")[8]. Indeed, we found that fitness defects under GdA treatment (50 μM) at 23 °C were correlated with the number of protein complexes ($\rho = 0.68$, $p = 0.025$, Spearman's rank correlation, Fig. 2e and Supplementary Fig. 4b). Moreover, we wanted to determine if protein complexes encoded by replaced chromosomes were associated with the formation of protein aggregates. Indeed, percentages of Hsp104 foci-containing cells in replacement lines at 180 min were also significantly correlated with chromosomal contributions to the formation of protein complexes ($\rho = 0.68$, $p = 0.03$, Spearman's rank correlation, Fig. 2f). It is interesting to note that the fitness defect is not directly proportional to the length of replaced chromosomes. For example, Chromosome 16 is shorter than Chromosome(s) 6+10, 7, 12, or 13, but 16L is more defective than 6+10L, 7L, 12L, or 13L. We also tested whether the fitness defect is correlated with the ratio of proteins in a complex divided by total proteins on the replaced chromosome. No significant correlation ($p > 0.05$) was observed between this ratio and fitness defects under GdA or Hsp104 foci. Moreover, several functions and processes related to proteostasis are specifically enriched on protein complex subunits encoded on Chromosomes 16, 8, and 15 (Supplementary Data 3). This suggests that the fitness defect is driven by the nature of protein complex subunits encoded on the replaced chromosome, rather than the number of genes on a chromosome.

## Multiple protein complexes are destabilized in the most defective replacement lines, 16L and 8+15L

Our genetic experiments and protein complex correlation analysis suggested that unstable chimeric protein complexes are the major cause of proteotoxic stress underlying complex incompatibilities. To directly test the "unstable complex" hypothesis, we characterized native protein complex formation in the most defective replacement lines, 8+15L and 16L, and compared it with the parental Sc strain. We predicted that the replacement lines should display more unstable

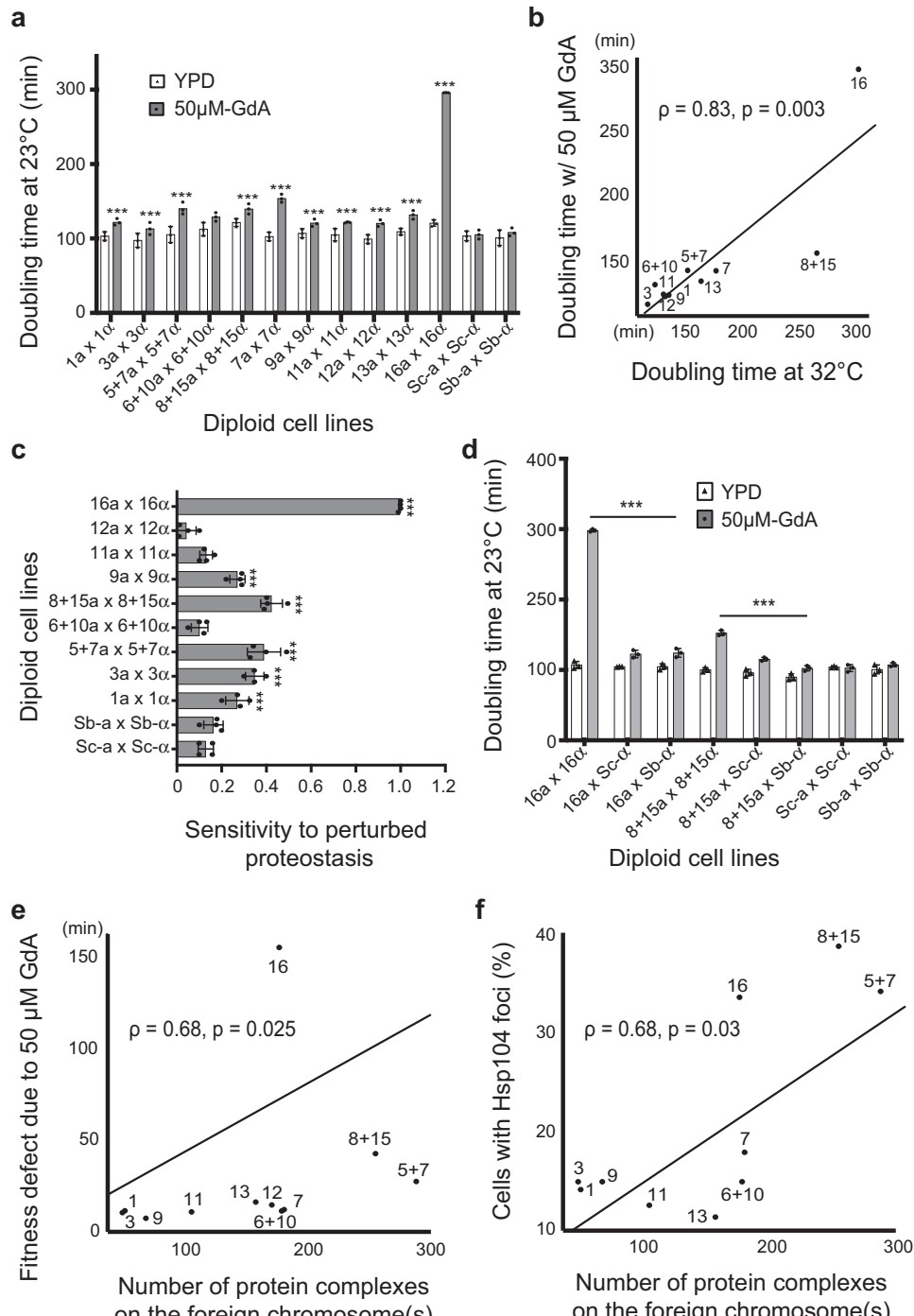

**Fig. 2 | Proteotoxic stress-induced incompatibility leads to mitotic and meiotic defects. a** Growth rates of diploid replacement lines are significantly reduced when Hsp90 is mildly compromised. The doubling time of *S. cerevisiae* (Sc), *S. bayanus* var *uvarum* (Sb), and replacement lines was measured and compared with or without the Hsp90 inhibitor, Geldanamycin (50 μM GdA), at 23 °C ($n = 3$). **b** The growth defect is not specific to compromised Hsp90. When replacement lines were grown in YPD at 32 °C, the doubling time is highly correlated with the data for cells treated with GdA ($n = 3$, Spearmans $\rho = 0.83$, $p = 0.03$). **c** Several replacement lines exhibit significant meiotic defects when pre-treated with a low dosage of GdA. Cells were grown in pre-sporulation medium with 50 μM GdA and then induced to sporulate under normal conditions without GdA ($n = 4$). The temperature was always maintained at 23 °C. Sensitivity was calculated as 1 - (sporulation frequency with GdA pre-treatment/sporulation frequency without GdA pre-treatment) and compared to both parents. **d**, Growth defects of the 16L and 8+15L lines can be partially rescued by providing a complete set of Sc or Sb genomes. The 16L and

8+15L haploid cells were mated with *S. cerevisiae* (16L × Sc and 8+15L × Sc) or *S. bayanus* var *uvarum* (Sb) (16L × Sb and 8+15L × Sb) to generate heterozygous diploid cells and their doubling time was measured in the presence of 50 μM GdA. The doubling time of homozygous diploid replacement lines was compared to that of the heterozygous diploids ($n = 3$). **e** The fitness defect under GdA treatment (50 μM) is significantly correlated with the number of complexes having subunits encoded on the replaced chromosomes (Spearmans $\rho = 0.68$, $p = 0.025$). **f** The percentages of cells containing Hsp104-mCherry foci are significantly correlated with the number of complexes having subunits encoded on the replaced chromosomes (Spearmans $\rho = 0.68$, $p = 0.03$). Hsp104 aggregates were counted in cells after having been shifted to 37 °C for 180 min ($n = 8$, $N \geq 500$ cells per time-point). The data are presented as mean values +/- SEM. ***: *p*-value < $10^{-3}$, one-sided Student's t-test. Source data and detailed statistical information are provided as a Source Data file.

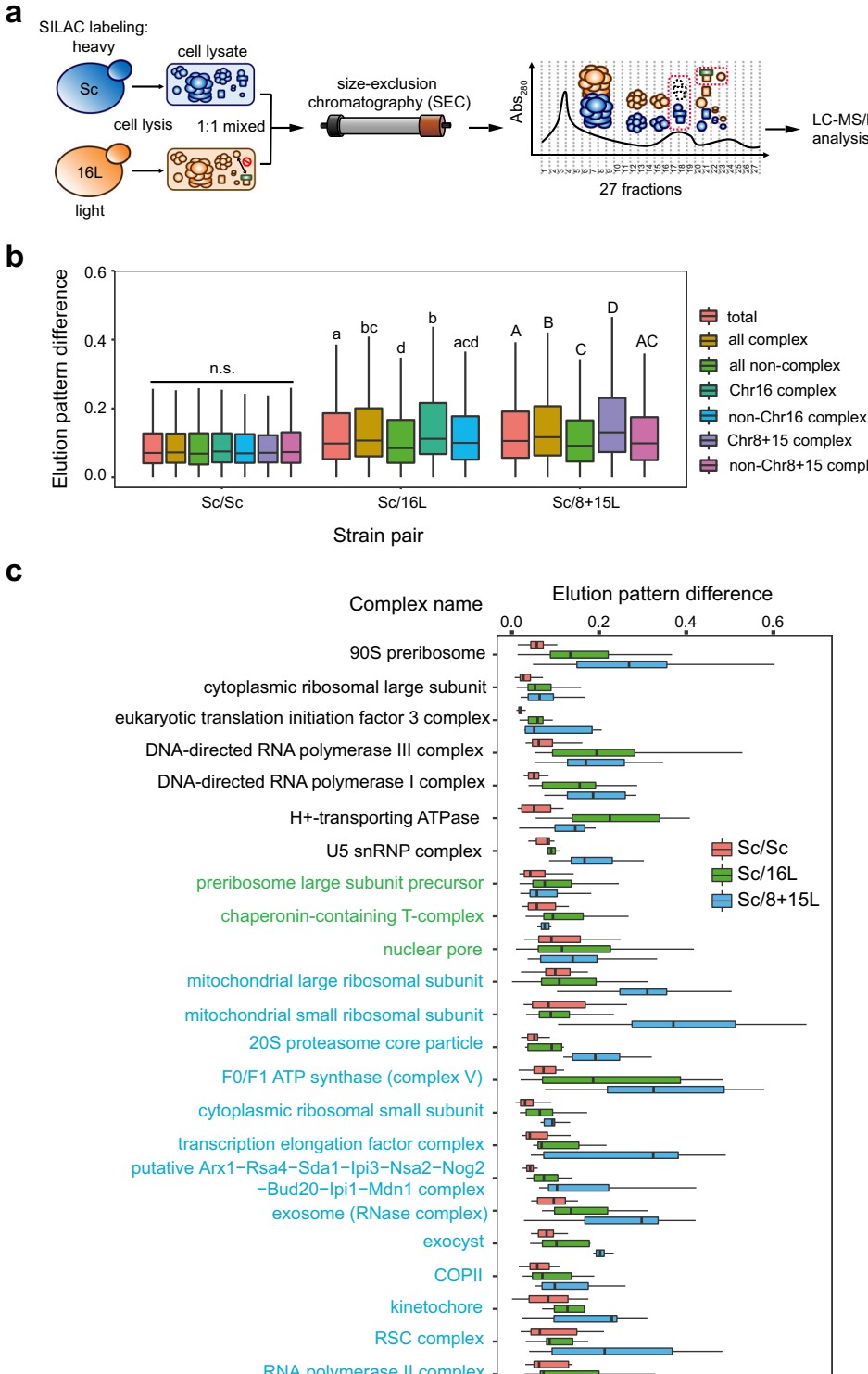

protein complexes than Sc and these would be enriched with subunits encoded by replaced Chromosomes 8, 15, and 16.

We monitored the formation of soluble protein complexes in yeast cells using native size-exclusion chromatography (SEC) followed by mass spectrometry (Fig. 3a, see Methods for details). The elution patterns of subunits in a protein complex through SEC represent the stability of the protein complex (Fig. 3a and Supplementary Fig. 3). For a stable protein complex, most subunits are expected to elute as a few continuous high-molecular-weight fractions. However, if the protein complex becomes unstable, the subunits dissociate and elute as low-

molecular-weight fractions, resulting in a different elution pattern (Fig. 3). To accurately quantify the elution pattern difference (EPD), we incorporated stable isotope labeling by amino acids in cell culture (SILAC) into our experimental procedures to enable an independent direct comparison of the top two defective replacement lines (8+15L and 16L) to the parental strain (Sc) (Fig. 3a and Supplementary Fig. 5).

We identified a total of 2432 proteins common to the experimental Sc-heavy/8+15L-light (Sc/8+15L, 3432 proteins), Sc-heavy/16L-light (Sc/16L, 2742 proteins) and control Sc-heavy/Sc-light (Sc/Sc, 2856 proteins) sets. Among them, 1359 proteins were subunits of 463

**Fig. 3 | Multiple protein complexes are destabilized in 8+15L and 16L cells.**
**a** Workflow of the SEC-based analysis of the formation of protein complexes. SEC, SILAC, and mass spectrometry were combined to compare the formation of protein complexes in Sc and 16L cells (see Methods for details). The dotted red boxes highlight a stable complex in the Sc line but partially disassembled in 16L cells. The subunits of this complex are expected to elute in different fractions for the Sc and 16L cells, leading to distinct elution patterns. **b** The protein complexes with subunits on Chromosomes 8+15 and Chromosome 16 are less stable than non-Chromosome 8+15 and 16 subunit-containing complexes. Elution pattern difference (EPD) values were calculated for every protein in Sc-heavy/Sc-light (Sc/Sc), Sc-heavy/16L-light (Sc/16L), and Sc-heavy/8+15L-light (Sc/8+15L) sets (also see Supplementary Fig. 5). The distributions of EPD values for different groups of proteins of the Sc/Sc, Sc/16L and Sc/8+15L sets are shown in box plots and the protein numbers of each group are 2342 (total), 1359 (complex), 983 (non-complex), 735

(Chr16 complex), 624 (non-Chr16 complex), 866 (Chr8+15 complex), and 493 (non-Chr8+15 complex). Distributions with the same letter (above each boxplot) are not significantly different from each other (Dunn's pairwise tests with Bonferroni correction, *p*-values > 0.05, see Supplementary Data 11 for the *p*-values of all Dunn's pairwise tests). n.s.: not significant. **c** Twenty-three protein complexes are significantly destabilized in 8+15L and 16L cells. A protein complex was defined as unstable if the EPD values of the complex subunits in the Sc/16L or Sc/8+15L sets were significantly higher than those in the Sc/Sc set. Here, 7 protein complexes are destabilized in both 8+15L and 16L cells (black), 3 unstable complexes are specific to 16L (green) and 13 complexes are unstable only in 8+15L (blue). Also, see Supplementary Fig. 6. Boxplots indicate median (middle line), 25th and 75th percentile (box), and min and max (whiskers). The summary of boxplots is provided in Supplementary Data 12 and source data are provided as a Source Data file.

previously identified protein complexes (Supplementary Data 4 and 5)[8]. The control Sc/Sc set did not display a significant difference in EPD values for complex and non-complex proteins, nor between any other groups (Fig. 3b; Kruskal–Wallis H test, $H(6) = 0.42$, $p = 0.65$), indicating that our experimental procedures were robust. In contrast, there was a significant difference in the EPD values between different groups in the experimental Sc/8+15L and Sc/16L sets (Fig. 3b; Kruskal–Wallis H test, 8+15L: $H(4) = 112.16$, $p = 2.2 \times 10^{-16}$ and 16L: $H(4) = 67.12$, $p = 9.2 \times 10^{-14}$). Pairwise tests between individual groups showed that the EPD values (median) of complex subunits were significantly higher than non-complex proteins (Dunn's pairwise tests with Bonferroni correction, adjusted *p*-values = $9.9 \times 10^{-12}$ (Sc/8+15L) and $5.9 \times 10^{-9}$ (Sc/16L)). To rule out the possibility that proteins encoded by Sb chromosomes might have different elution patterns contributing to observed EPDs, EPD values were recalculated after removing the proteins encoded by replaced chromosomes. We observed similar results (Supplementary Fig. 6). These data provide direct evidence that many protein complexes had become destabilized in 8+15L and 16L cells.

If the complex instability is due to incompatibility between *S. cerevisiae* and *S. bayanus* var. *uvarum* protein subunits (i.e., species-specific interactions), we anticipated that the complexes having subunits encoded by foreign Chromosomes 8, 15, and 16 should be less stable than those lacking such subunits. Indeed, Chr8+15 and Chr16 complex proteins (i.e., proteins from the complexes containing subunits from Chromosomes 8+15 and 16, respectively) presented significantly higher EPD values compared to non-Chr8+15 and non-Chr16 complex proteins (i.e., proteins from the complexes lacking any Chromosome8+15-encoded and 16-encoded subunit, respectively) (adjusted p-values = $1.0 \times 10^{-10}$ (Sc/8+15L) and $3.1 \times 10^{-4}$ (Sc/16L), Fig. 3b, Supplementary Fig. 6a, b). Furthermore, there was no significant difference between the EPD values of non-Chr8+15 and non-Chr16 complexes and non-complex proteins, respectively (adjusted *p*-values = 1 (Sc/8+15L) and 0.06 (Sc/16L)). Thus, the presence of foreign complex subunits was likely the primary contributory factor for the instability of protein complexes in 8+15L and 16L cells.

In order to understand the general features of complex incompatibility, we endeavored to identify protein complexes that were destabilized as a whole rather than in few individual subunits. A protein complex was deemed unstable in the 8+15L or 16L cells if the EPD values of the complex subunits in the Sc/8+15L or Sc/16L sets were significantly higher than those in the Sc/Sc set (Wilcoxon signed-rank tests, Benjamini–Hochberg FDR corrected *p*-values < 0.05, see "Methods" for details). While 20 complexes were unstable in Sc/8+15L, 10 protein complexes were unstable in Sc/16L (Fig. 3c and Supplementary Data 6a). Among them, complexes with subunits encoded by the replaced chromosomes (i.e., the chimeric protein complexes) were prone to be destabilized (9 out of 10 in 16L and 19 out of 20 in 8+15L, *p*-value < 0.05, Fisher's exact test, Supplementary Data 6a). We also performed the complex stability analysis after removing the complex

subunits encoded by replaced chromosomes and observed similar results (Supplementary Fig. 6c, d and Supplementary Data 6b). Interestingly, most of the unstable chimeric complexes are involved in basic cellular functions, including transcription, translation, and respiration. These data further support that species-specific interactions between complex subunits can evolve rapidly, even within complexes having essential cellular functions.

## Unstable chimeric protein complexes have lower soluble protein abundances in 8+15L and 16L cells

Disassembly of protein complexes can result in degradation or insoluble aggregate formation of dissociated protein subunits[50,51]. If cells contain many unstable protein complexes, their systems governing proteostasis may be overwhelmed and further compromised. We examined if there was a decrease in the abundance of the destabilized protein complex subunits in 8+15L and 16L cells. Further analysis of our SILAC data revealed significantly lower levels of complex subunit proteins compared to non-complex proteins in 8+15L and 16L cells (Fig. 4a and Supplementary Fig. 7). Consistent with our EPD data, the subunit abundance of Chr8+15 and Chr16 complexes was significantly lower than for non-Chr8+15 and non-Chr16 complexes (Fig. 4a, Supplementary Fig. 7, and Supplementary Data 7).

Since the replaced chromosomes also impacted global gene expression (Fig. 1a and Supplementary Data 1), we tested whether the observed reduction of complex subunit abundance was simply due to decreased gene expression by computing the Spearman's correlation coefficient between the protein (SILAC-ratios) and transcript levels. We found that the protein and transcript levels were least correlated in both Chr16 and Chr8+15 complex groups compared to the other groups (Supplementary Data 8). Fisher's r-to-z transformation, which can compare the significance in the difference between correlation coefficients[52], also showed that the Spearman's correlation coefficients in the Chr8+15 complex group ($\rho = 0.24$) and the Chr16 complex group ($\rho = 0.19$) were significantly lower than that of the non-Chr8+15 complex group ($\rho = 0.35$) and non-Chr16 complex group ($\rho = 0.45$), or the non-complex group (8+15L: $\rho = 0.39$, 16L: $\rho = 0.46$) (*p*-values < 0.05). On the other hand, we did not find the correlation coefficients for the non-complex and non-Chr8+15 complex groups or non-complex and non-Chr16 complex groups respectively to be significantly different (*p*-value = 0.31). These data suggest that gene expression change is not the only factor contributing to the observed reduction of subunit protein abundance and translational or post-translational regulation is involved.

To further validate our proteomics data, we examined one of the unstable complexes, RNA polymerase III, which was identified in both 8+15L and 16L and could be biochemically purified from total cell extracts[53]. We calculated a close to two-fold reduction (49%) in the abundance of RNA polymerase III subunits in 16L cells, even though the same total amounts of protein lysates from 16L and Sc cells were used

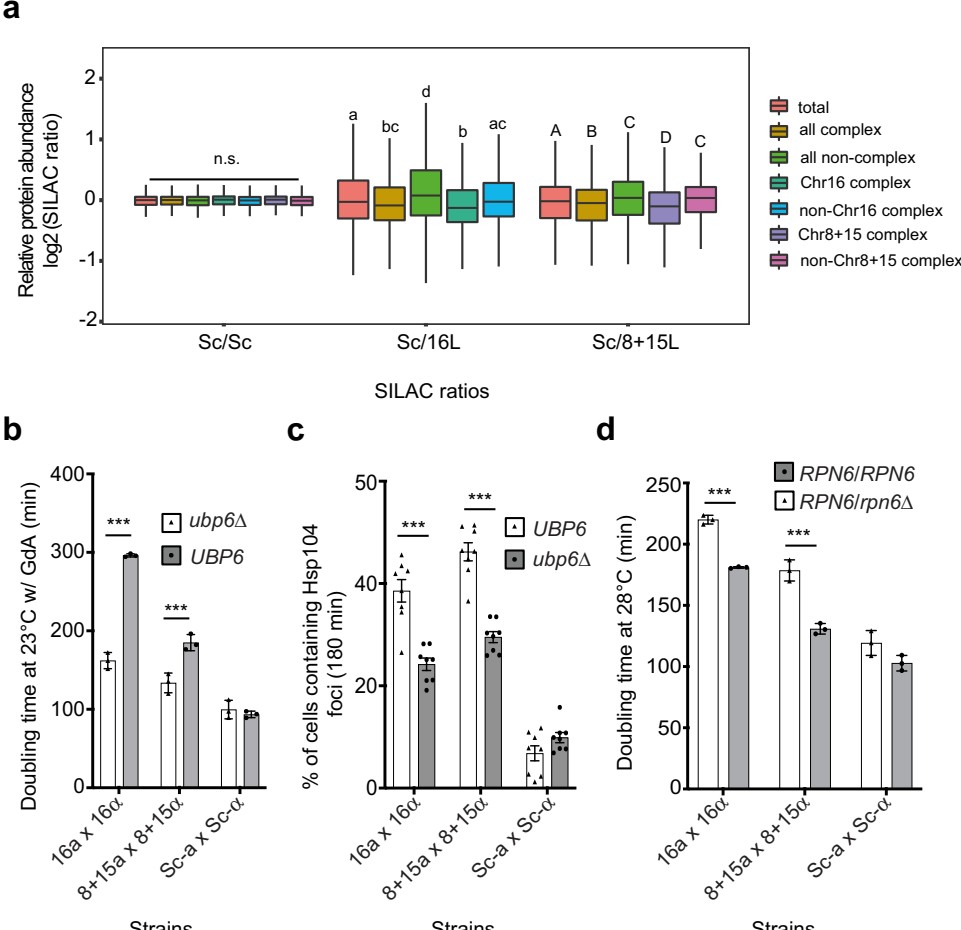

**Fig. 4 | Proteotoxic stress in the replacement lines can be partially relieved by up-regulating the protein degradation machinery. a**, Protein abundances of complexes having subunits encoded on Chromosome 16 are significantly reduced in 16L cells and Chromosomes 8+15 are reduced in 8+15L cells when compared to non-Chr16/non-Chr8+15 complexes, respectively (also see Supplementary Fig. 7). The distributions of log2-transformed SILAC ratios of proteins in different groups are shown in boxplots. Distributions with the same letter (above each boxplot) are not significantly different from each other (Dunn's pairwise tests with Bonferroni correction, *p* > 0.05, see Supplementary Data 11 for the *p*-values of all Dunn's pairwise tests). n.s.: not significant. **b** Growth defects of the 16L and 8+15L lines are partially rescued by deleting the *UBP6* gene. The *ubp6*Δ mutants exhibit enhanced proteasomal degradation activity, thereby facilitating the removal of destabilized complex subunits. Diploid cell lines were grown in YPD with 50 μM Geldanamycin

(GdA) at 23 °C and their doubling times were compared (*n* = 3). **c** Protein aggregate load of the 16L and 8+15L lines is alleviated in *ubp6*Δ mutants. The Hsp104 aggregate data were obtained after the cells had been shifted from 23 to 37 °C for 180 min (*n* = 8; SEM, *N* ≥ 500 cells per time-point). **d** Growth defects of the 16L and 8+15L lines are aggravated in the heterozygous *RPN6/rpn6*Δ mutants. Rpn6 is an essential component of proteasomes, and proteasomal degradation activity is mildly compromised in *RPN6/rpn6*Δ mutants. Diploid cell lines were grown in YPD at 28 °C and their doubling times were compared (*n* = 3). The data are presented as mean values +/− SEM. ***: *p*-value < $10^{-3}$, one-sided Student's t-test. Boxplots indicate median (middle line), 25th and 75th percentile (box), and min and max (whiskers). The summary of boxplots is provided in Supplementary Data 12. Source data and detailed statistical information are provided as a Source Data file.

for the pull-down experiment (Supplementary Fig. 8a). Together, these data indicate that once subunits have dissociated from destabilized complex, they are quickly degraded or form insoluble aggregates.

## Up-regulating the protein degradation machinery alleviates the fitness defects of replacement lines

Since individual incompatible loci only contribute to a small proportion of the fitness defects, it is almost impossible to confirm their effect specifically. Instead, we tested whether the fitness defect could be relieved by downregulating global proteotoxicity. The ubiquitin-proteasome machinery is a primary pathway known to regulate unbalanced multi-protein complexes. Cells exhibit intrinsic proteotoxic stress when proteasomes are overwhelmed by extensive perturbations[51,54]. In ubiquitin-mediated proteasomal degradation, multiple ubiquitin molecules are covalently linked to candidate substrates and act as recognition motifs for 26S proteasomes[55]. Ubp6, a ubiquitin-specific protease, has a dual role in ubiquitin recycling and

regulation of proteasomal degradation. Proteasomal degradation activity is accelerated in the absence of Ubp6[56–58]. We postulated that if excess amounts of destabilized complex proteins in the replacement lines overburdened proteasomes to induce fitness defects, an absence of *UBP6* should allow the proteasomes to degrade these proteins more efficiently and improve cell proliferation by the replacement lines. We used *ubp6*Δ mutants of the 8+15L and 16L lines exhibiting the greatest proliferative defects to perform growth assays under the treatment of 50 μM GdA at 23 °C. As anticipated, the fitness defect was indeed rescued in our *ubp6*Δ mutant replacement lines (Fig. 4b), suggesting that proteotoxic stress due to overburdening of proteasomes was responsible for the growth defects. In addition, we performed protein aggregation assays in both mutant lines and observed that the number of cells harboring Hsp104-mCherry foci decreased significantly in the *ubp6*Δ mutants (Fig. 4c and Supplementary Fig. 8b), further confirming that *ubp6* deletion partially relieved intrinsic proteotoxic stress in the 8+15L and 16L cells.

If 26S proteasomes are the most crucial degradation machinery controlling protein complex homeostasis in our replacement lines, we expected hybrid cells to display severe growth defects when their proteasomes are compromised, even under normal growth conditions. We constructed heterozygous deletion mutants of Rpn6, an essential lid component of the 26S proteasome, from our 8+15L and 16L lines, and measured their fitness at 23 and 28 °C. Heterozygous *RPN6/rpn6Δ* mutation only partially compromised the activity of proteasomes and had mild impacts on the fitness of Sc cells at both temperatures. In contrast, the *RPN6/rpn6Δ* mutants of the 16L and 8+15L lines revealed a significant fitness defect at 28 °C (Fig. 4d and Supplementary Fig. 8c).

Lastly, we confirmed that the proteotoxic stress observed in the replacement line was not due to an ineffective proteasome system. Proteasomal activity assays showed that the activity of endogenous 26S proteasomes in 16L cells did not differ from that of Sc cells (Supplementary Fig. 8d). Together, our results demonstrate that destabilized protein complexes in hybrid cells often increase the burden of proteasomes, a key regulator of proteostasis. Depending on the number and abundance of chimeric complexes, that overburdening results in differential levels of hybrid incompatibility.

## Discussion

The mechanisms underlying speciation are a long-standing mystery in evolutionary biology. Over the past three decades, scientists have discovered dozens of "speciation genes", mainly involving two interacting components best described by the simplest form of Dobzhansky–Muller incompatibilities[17,31,59]. These findings allow us to gradually depict the possible driving forces behind speciation. However, our knowledge remains limited of the molecular mechanisms underlying complex incompatibilities (i.e., involving more than two genetic loci), which are widely observed[18–20] and believed to be crucial during incipient speciation[21,22]. By using different chromosome replacement lines, our experiments have revealed that the proteotoxicity caused by destabilized chimeric protein complexes may represent a general mechanism of complex incompatibility.

Approximately half of the *S. cerevisiae* proteome is known to constitute protein complex subunits[6,8], and the proteomes of other eukaryotic organisms are probably also composed of similar proportions of protein complexes[60]. The functions of protein complexes are often deemed evolutionarily conserved, especially for those involved in basic cellular pathways[61]. So why does incompatibility evolve between the complex subunits of closely related species? One plausible hypothesis is that the complex structure provides a microenvironment allowing its subunits to drift and coevolve. The stability of a protein complex is maintained by the interactions between multiple protein interfaces. As long as the whole complex is stably assembled to execute its function, individual interactions between subunits may change either by neutral mutations or by adaptation. Ultimately, some complex subunits may exhibit distinct evolutionary trajectories among different lineages, resulting in incompatibility when hybrids are formed[23]. This idea is corroborated by two studies that analyzed protein-protein interactions (PPIs) in protein complexes using the F1 hybrid diploids of different yeast species[62,63]. While many PPIs are conserved in hybrids, interactions involved in some biological functions related to proteostasis, metabolism, and mitochondria are specifically altered.

It is worth noting that our chromosome replacement lines are more similar to the F1 haploid gametes or homozygous F2 progeny of hybrids that only carry one parental allele of each gene. In the F1 hybrid diploids, both parental alleles are still present that reduces the proportion of chimeric protein complex formation. More importantly, incompatible subunits will have a chance to find their authentic interacting partners that are unavailable in the hybrid progeny carrying only one parental allele of each gene. Consistent with this idea, F1 hybrid

diploids of *S. cerevisiae* and *S. bayanus* var *uvarum* had less severe proteotoxic stress (Supplementary Fig. 2) and did not show growth defects in the laboratory condition. In contrast, both F1 gametes and F2 homozygous progeny suffered from obvious defects[28,64].

Maintaining protein homeostasis is one of the most challenging tasks for organisms living in constantly changing natural habitats. Cells have developed an extensive protective network to deal with the proteotoxicity caused by various environmental stresses[2]. However, that system of proteostasis may still be overwhelmed upon encountering large-scale genomic alteration. For example, in the presence of even just one additional chromosome, aneuploid cells suffer severe proteotoxic stress[40,65,66]. Proteotoxicity in aneuploids is mainly attributed to the stoichiometrically imbalanced proteome due to extra chromosomal copies overburdening the proteostasis machinery[41,67]. However, in hybrids between related species in which synteny and orthology are chiefly conserved, the source of proteotoxicity is more complicated.

Similar to our results, a proteomics study in *Drosophila* hybrids also showed that proteins involved in proteostasis were significantly upregulated in developing hybrids[68]. The observed proteotoxicity in hybrid cells can be due to multiple reasons. First, a mis-regulated transcriptional network in hybrid cells can lead to altered gene expression, causing stoichiometrically imbalanced complex subunits, similar to aneuploid cells. Nonetheless, our correlation analysis indicates that the mRNA and protein levels are not well correlated especially for those complexes significantly influenced by replaced chromosomes, suggesting that translational or post-translational regulation is also involved. Second, genetic divergence between two parental backgrounds can affect the relationship of interacting proteins by changing the partners or altering the strength of interactions. It will result in assembling failure or unstable chimeric complexes depending on the level of divergence. Since many complexes are involved in protein homeostasis, the effect can be cumulative. Third, an excess amount of dissociated subunits can further interfere with protein homeostasis. Although cells possess specific machineries to degrade dissociated subunits and prevent the formation of insoluble aggregates, these machineries can become overwhelmed and saturated when the parental genomes have diverged beyond a critical level.

Our results suggest that intrinsic proteotoxic stress likely represents a general defect of different hybrid genomes since it arises from the combined action of many genes and does not require specific mutations with strong effects. Moreover, the stability of a protein complex is often determined by interactions between multiple subunits. Individually the incompatible subunits may only have negligible effects, but can be synergistic when some or all the incompatible subunits are present, resulting in epistasis observed in complex incompatibilities (Supplementary Fig. 9)[69]. This defect can arise at an early stage of speciation and continuously build up to achieve complete reproductive isolation[15,70].

## Methods

### Yeast strains

The parental *S. cerevisiae MATa* and *MATα* strains (JYL1127 and JYL1128) are isogenic with W303 (*ura3-1 his3-11,15 leu2-3,112 trp1-1 ade2-1 can1-100*). The parental *S. bayanus MATa* and *MATα* strains (JYL1030 and JYL1031*, renamed as *S. bayanus* var *uvarum*) were derived from a strain (*S. bayanus* #180) collected by Dr. Duccio Cavalieri (University of Florence, Italy). Construction of the chromosome replacement lines was reported in Lee et al.[28]. The vegetative lifecycle of the wild yeast *S. cerevisiae* is predominantly diploid and, under stress conditions, budding yeasts tend to exhibit diploid superiority[71,72]. Moreover, diploid cells can be used to ascertain the effect of foreign chromosomes during meiosis. Thus, for each of the eleven replacement lines, *a*- and *α*-types of each isogenic replacement line were systematically

crossed to obtain homozygous diploid replacement lines. We used polymerase chain reaction (PCR) to amplify the deletion cassettes of deletion mutants from the yeast deletion collection[73]. Genes were then deleted by transforming the purified DNA fragment of deletion cassettes into the strain of interest and selecting transformants. For the SILAC experiment, *LYS2, CAR2*, and *ARG4* were deleted in the *S. cerevisiae* pure strain and 16L replacement line. We replaced the *can1-100* allele with the wild-type *CAN1* allele in order to delete *ARG4* in the W303 background.

To construct the Hsp104-mCherry-carrying strains, the promoter and coding sequences of *S. cerevisiae HSP104* were fused with the mCherry fluorescent protein and a *Kluyveromyces lactis URA3* marker using the three-fragment PCR method. The constructed fragment was then inserted into the endogenous *HSP104* locus. The 12L line was excluded from the experiment as *HSP104* is located on this chromosome. Details of the primers used to create deletion strains are presented in Supplementary Data 9. Details of the strains used in this study are provided in Supplementary Data 10. $SSD1^{YPS1009}$ plus 1000 bp upstream and 300 bp downstream regions were cloned into the CEN plasmid pRS41N for complementation in Sc, 8+15L and 16L (both with and without Hsp104-mCherry containing strains).

## RNA sequencing and data analysis
We extracted mRNA from log-phase cells ($OD_{600} < 0.8$) cultured in YPD (1% yeast extract, 2% bactopeptone, and 2% glucose) at 23 °C by the phenol-chloroform method[74]. The quality and quantity of RNA samples were measured using an Agilent 2100 Bioanalyzer (Agilent, Santa Clara, CA, USA) and the Qubit assay (Thermo Fisher Scientific, Bremen, Germany). Libraries were prepared by Welgene Biotech (Taipei, Taiwan) using the Agilent SureSelect stranded RNA library preparation kit (Agilent) on an Agilent Bravo liquid handling system (Agilent). The libraries were single-end-sequenced (75 base pairs, bp) on an Illumina NextSeq500 instrument (Illumina, San Diego, CA, USA).

RNA sequencing data from different replacement lines were trimmed using Trimmomatic (version 0.38) and mapped to the corresponding transcriptome using Salmon (version 0.12.0)[75,76]. The read counts (the "NumReads" column in Salmon output files) of genes from *S. cerevisiae* chromosomes were analyzed using DESeq2 (version 1.22.2) with default settings to calculate fold-change normalized to the pure *S. cerevisiae* strain[77]. A gene was considered differentially expressed if fold-change >1.5 and adjusted *p*-value < 0.05. Genes that were commonly up- or downregulated in at least four replacement lines were defined as common-response genes (Supplementary Data 1). However, if the gene was differentially expressed in both directions in at least four lines, it was excluded from the common-response gene list. We performed the correlation analysis by calculating Spearman's correlation coefficients between the expression levels of ESR genes in replacement lines (median values of log2 fold-change among all replacement lines were used) and those under stress conditions[33,42] (Supplementary Data 2).

## Analysis of endogenous protein aggregates
Replacement lines carrying Hsp104 tagged with mCherry were grown in YPD medium at 23 °C to log-phase. The cell cultures were separated into two tubes and one of the tubes was shifted to 37 °C. Cells subjected to both temperatures were collected at various time points, diluted with PBS, and loaded into a glass-bottomed viewPlate-96F (PerkinElmer, Waltham, MA, USA) coated with concanavalin A (C2020, Sigma-Aldrich, St Louis, MO, USA). The plates were then centrifuged to attach the cells to the bottom of the plates, and images were immediately obtained using the ImageXpress MicroXL system (Molecular Device, Sunnyvale, CA, USA). All images were analyzed manually using ImageJ (http://rsbweb.nih.gov/ij). For each time-point, Hsp104-mCherry foci were counted from five fields of at least 500 cells.

## Growth curves and quantification
All growth assay experiments were conducted in triplicate in YPD media using log-phase cells in a Tecan F200 plate reader (Tecan, Mannedorf, Switzerland) without shaking. Measurements of absorbance at $A_{595nm}$ were made every 10 min. Maximum growth rates were calculated[78]. For Hsp90 inhibitor treatments, cells were grown in the medium containing 50 μM Geldanamycin and the growth rates were compared with that of untreated cells at 23 °C. For heat-stress assays, growth rates were measured at 32 °C.

## Sporulation assay
Homozygous diploid replacement lines were grown overnight in pre-sporulation medium (1% yeast extract, 2% bactopeptone, and 2% potassium acetate) with 50 μM Geldanamycin, washed, and then sporulated in 2% potassium acetate. Sensitivity was calculated using the formula, sensitivity = 1 − (sporulation frequency with GdA pretreatment/sporulation frequency without GdA). Sporulation frequency was obtained by counting at least 500 cells from sporulation cultures.

## Complex number correlation
The list of yeast protein complexes was downloaded from Costanzo, M. et al. [8], a recent compendium of yeast protein complexes. This list was manually inspected for physical protein-protein interactions and modified to remove genetic interactions and redundant protein complexes. The dataset used in this study consisted of 575 protein complexes and 2312 subunits (Supplementary Data 4). Details of modifications are provided in the same table. The fitness defect of each replacement line was calculated as the difference between the doubling times of replacement lines treated with 50 μM Geldanamycin at 23 °C (or without Geldanamycin at 32 °C) and the doubling times of corresponding replacement lines without any treatment at 23 °C. Spearman's correlation coefficient was calculated between the number of protein complexes that comprise subunits encoded by the replacement chromosomes and fitness defects under 50 μM Geldanamycin, under 32 °C, and the percentage of cells harboring Hsp104-mCherry foci.

## Growth conditions for SILAC labeling and lysate preparation for size-exclusion chromatography
Yeast cells were grown overnight to log phase ($OD_{600} = 0.5$) at 23 °C in synthetic medium containing 6.7 g/l yeast nitrogen base, 2% glucose, 80 mg/l each of L-alanine, L-asparagine, L-aspartic acid, L-cysteine, L-glutamic acid, L-glutamine, L-glycine, L-histidine, L-isoleucine, L-methionine, L-phenylalanine, L-proline, L-serine, L-threonine, L-tryptophan, L-tyrosine, L-valine, inositol, and uracil, 8 mg/l p-aminobenzoic acid, 400 mg/l L-leucine, and 20 mg/l adenine with the addition of either 20 mg/l heavy L-arginine (Arg, $^{13}C_6$, $^{15}N_4$, Thermo Fisher Scientific, Waltham, MA, USA) plus 30 mg/l heavy L-lysine (Lys, $^{13}C_6$, $^{15}N_2$; Cambridge Isotope Laboratories, Andover, MA, USA) or 20 mg/l light Arg plus 30 mg/l light Lys (Sigma-Aldrich). Then the cells were harvested, washed once with distilled water, and the cell pellets were kept at −80 °C before lysis. In order to compare the difference between Sc, 8+15L and 16L cells, the pure Sc strain was grown in the heavy medium and 8+15L and 16L cells were grown in the light medium. A pure Sc strain grown in the light medium was used as a control. The cell pellets were resuspended in lysis buffer (20 mM HEPES-KOH pH 7.4, 120 mM KCl, 2 mM EDTA, 0.5 mM DTT, 10% glycerol) with 1 mM PMSF and protease inhibitor cocktail set IV (Merck, Darmstadt, Germany). The cells were frozen in liquid nitrogen and then lysed using a 6875 Freezer/Mill High Capacity Cryogenic Grinder (SPEX SamplePrep, Metuchen, NJ, USA). The lysate was thawed and cleared by centrifugation at 15,000 × g for 15 min at 4 °C, and then filtered using 0.45 μm syringe filters (Minisart NML Syringe Filter 16555 K, Sartorius, Goettingen, Germany). The protein concentration of the cleared lysate was measured by the Bradford assay.

## Size-exclusion chromatography (SEC)

The same amounts of protein lysates from the pure Sc strain (labeled with heavy Arg and Lys) and the 8+15L or 16L line (labeled with light Arg and Lys) were mixed, and then a total of 200 μl of lysate was injected into a Superose 6 10/300GL column (GE Life Sciences, Chicago, IL, USA) equilibrated with the lysis buffer on an ÄKTA Purifier system (GE Life Sciences). The flow rate was 0.2 ml/min, and 81 200 μl fractions were collected. We pooled every three sequential fractions into one new fraction, resulting in a total of 27 fractions. For the control experiment, the same protein amounts of lysates from a pure Sc cell culture labeled with heavy Arg and Lys and another pure Sc cell culture labeled with light Arg and Lys were mixed and analyzed according to an identical protocol.

## Sample preparation for LC-MS/MS analysis

Proteins in the 27 SEC fractions (see above) were denatured by adding urea to a final concentration of 8 M, followed by reduction with 5 mM dithioerythritol at 37 °C for 45 min, and cysteine alkylation with 25 mM iodoacetamide at room temperature in the dark for 1 h. Protein samples were transferred to Amicon Ultra-0.5 centrifugal filters (10 kDa, Millipore, Burlington, MA, USA) and centrifuged at $13,200 \times g$ for 20 min. Buffer exchange was performed in two successive washes with 8 M urea in 25 mM HEPES pH 7.4. Protein concentrations were then determined by the Bradford assay. Samples were digested overnight at 37 °C using LysC protease and trypsin at an enzyme-to-substrate ratio of 1:50 (w/w). The total peptide concentration was measured via Pierce quantitative colorimetric peptide assays (Thermo Fisher Scientific). Peptide desalting was achieved using $C_{18}$ Stage Tips (Thermo Fisher Scientific), and 0.5 μg of the peptide from each sample was taken for LC-MS/MS analysis.

## LC-MS/MS analysis

NanoLC-nanoESi-MS/MS analysis was performed on a Thermo Ulti-Mate 3000 RSLCnano system connected to a Thermo Orbitrap Fusion mass spectrometer (Thermo Fisher Scientific) equipped with a nanospray interface (New Objective, Woburn, MA, USA). Peptide mixtures were loaded onto a 75 μm ID, 25 cm length PepMap $C_{18}$ column (Thermo Fisher Scientific) packed with 2 μm particles having a pore width of 100 Å and they were separated for 150 min using a segmented gradient from 5% to 35% solvent B (0.1% formic acid in acetonitrile) at a flow rate of 300 nl/min. Solvent A was 0.1% formic acid in water. The mass spectrometer was operated in the data-dependent mode. Briefly, survey scans of peptide precursors from 350 to 1600 $m/z$ were performed at a 120 K resolution with a $2 \times 10^5$ ion count target. Tandem MS was performed by isolation window at 2 Da with the quadrupole, HCD fragmentation with a normalized collision energy of 30, and rapid scan MS analysis in the ion trap. The MS2 ion count target was set to $10^4$ and the max injection time was 50 ms. Only precursors with charge states of 2–6 were sampled for MS2. The instrument was run in top speed mode with 3 s cycles, and the dynamic exclusion duration was set to 60 s with a 10 ppm tolerance around the selected precursor and its isotopes. Monoisotopic precursor selection was turned on.

## Data analysis of LC-MS/MS results and identification of unstable protein complexes in replacement lines, 8+15L and 16L

A custom database of yeast protein sequences was built for the SILAC comparison of the proteins encoded on Chromosome 16. Briefly, the protein sequences of Chromosome 16 from *S. cerevisiae* were compared to those from *S. bayanus* var *uvarum*, and the common trypsin-digested peptide sequences were used for the SILAC quantification. Mass spectrometry data were processed with MaxQuant software[79] (v. 1.6.7.0) according to a protocol described previously[80]. Peptides and proteins were identified using the Andromeda search engine against the custom yeast database with the following search parameters:

carbamidomethylation of cysteine as a fixed modification; oxidation of methionine, deamidation of asparagine and glutamine, acetylation of protein N-termini, and trypsin cleavage with a maximum of two missed cleavages. For analysis of the control set, the pure Sc protein sequence database was used. The unique peptides were used for peptide quantification. To improve the number of peptides that could be used for protein quantification and relative abundance profiling across SEC fractions, the match between runs option was enabled with a matching window set to 0.7 min and an alignment window of 20 min. The re-quantify option was also enabled. The false discovery rate (FDR) of peptides and protein identification was set at 1%. All other MaxQuant parameters were left as the default options. All protein identifications were required to have at least one unique peptide.

For relative protein quantification, the extract ion current intensities reported by MaxQuant were used. The intensities in different fractions were corrected by multiplying by the peptide concentrations in the corresponding fractions. To avoid the potential effect of differences in protein levels for different strains, the heavy and light intensities of a protein were first divided by the sum of the heavy intensities and the sum of the light intensities, respectively, for conversion into percentages in different fractions, which we define as the elution pattern. The elution pattern of a protein reflected the status of protein interactions (Supplementary Fig. 3). The intensities and elution patterns (percentage in every fraction) of all proteins in experimental and control sets are presented in Supplementary Data 5. If protein complexes were unstable and disassembled in the replacement line, then the elution patterns of the protein components in the replacement line would deviate from those in the pure Sc strain. The sums of absolute values of the difference between the heavy and light percentages from all 27 fractions were used to quantify the EPD between Sc and 8+15L or 16L proteins (EPD = $\sum_{f=1}^{27} |d_f|$; $d$ = percentage difference, $f$ = fraction number). The database of protein complexes determined based on physical protein-protein interactions was adopted from a previous study[8] and modified to remove genetic interactions and several redundant complexes (see Supplementary Data 4 for details).

To determine if a protein complex was unstable in 8+15L or 16L cells, the EPD values of all the protein components in the complex were compared to those from the control experiment using Wilcoxon signed-rank tests, and *p*-values were corrected using the Benjamini–Hochberg procedure with a FDR of 0.05[81]. Eighty-one protein complexes with enough protein components for statistical analysis are shown in Supplementary Data 6.

To determine if a protein complex has reduced protein abundance in 8+15L or 16L cells, the SILAC ratios of all the protein components in the complex were compared to those from the control experiment using Wilcoxon signed-rank tests, and *p*-values were corrected using the Benjamini–Hochberg procedure with a FDR of 0.05[81]. Eighty-one protein complexes with enough protein components for statistical analysis are shown in Supplementary Data 7.

## Purification of RNA polymerase III

RNA polymerase III was purified based on a protocol described previously[53], but with the following modifications. Ret1 was tagged with the C-terminal TAP-tag[82] as bait to pull down RNA polymerase III. A total of 5 liters of yeast cell culture was grown in YPD medium to an $OD_{600}$ of 1. The harvested cells were resuspended in the purification buffer, frozen in liquid nitrogen, and then lysed using a 6875 Freezer/Mill High Capacity Cryogenic Grinder (SPEX Sample-Prep). The NaCl concentration in the wash buffer was elevated to 1 M. After TEV protease digestion, three volumes of the supernatant were mixed with one volume of 4X SDS-loading buffer and boiled for 10 min. Purified RNA polymerase III in the supernatant was separated using SDS-PAGE gels, and the gels were stained with EBL Easy Blue-Plus (EBL, New Taipei City, Taiwan) to visualize proteins.

The gel was scanned and intensities of different bands were quantified using ImageJ (https://imagej.nih.gov/ij/).

## Identification of the components of RNA polymerase III by mass spectrometry

After the staining procedure, gel bands were excised and cut into small pieces. A modified in-gel digestion protocol was applied[83]. Briefly, after sequentially washing the gel pieces with 25 mM $NH_4HCO_3$, 40% methanol solution, and 100% acetonitrile, DTT reduction and alkylation with iodoacetamide of the proteins in gel pieces were performed. The gel pieces were washed and dried in a vacuum centrifuge before trypsin digestion. A trypsin solution of 25–30 µl 25 mM $NH_4HCO_3$ containing 75–100 ng of sequencing-grade modified trypsin (Promega, Madison, WI, USA) was added to gel pieces and incubated for 12–16 h at 37 °C. The reaction was stopped by adding 1–2 µl of 5% formic acid. The digested samples (0.5 µl) were carefully mixed with 0.5 µl matrix solution and 0.5 µl of the mixture was deposited onto the 384/600 µm MTP AnchorChip (Bruker Daltonics, Billerica, MA, USA). All mass spectrometry experiments were done using a Bruker Autoflex III MALDI TOF/TOF mass spectrometer (Bruker Daltonics) equipped with a 200 Hz SmartBean Laser in positive ion mode with delayed extraction in the reflectron mode. Data acquisition was done manually with Flex-Control 3.4, and data processing was performed using Flex-Analysis 3.4 (both Bruker Daltonics). Protein database searches, through Mascot, using combined PMF and MS/MS datasets were performed via Biotools 3.2 (Bruker Daltonics).

## Proteasome activity assay

The in-gel proteasome activity assay was performed based on the protocol of a previous study[84], with the following modifications. Briefly, the cells were grown at 23 °C to log phase ($OD_{600}$ = 0.4–0.6) and harvested. The harvested cell pellet was resuspended in buffer A (50 mM Tris-Cl pH 7.4, 5 mM $MgCl_2$, 10% glycerol, 1 mM DTT) and immediately lysed by mechanical disruption with glass beads. The cell lysate was cleared by centrifugation at 15,000 × $g$ for 20 min and then at 100,000 × $g$ for 30 min at 4 °C. The protein concentration was determined using the Bradford assay. One hundred µg of the cleared cell lysate from each sample was resolved by nondenaturing PAGE[85] in order to separate doubly-capped 26S (RP2CP), singly-capped 26S (RP1CP), and free 20S (CP) proteasomes. The gels were then incubated for 20 min at 37 °C in 10 ml of buffer A with 0.1 mM Suc-LLVY-AMC (S6510, Sigma-Aldrich) and 1 mM ATP. Signals of 26S proteasome activity were measured upon exposure to UV light using a UVP BioSpectrum 815 system equipped with a FirstLight UV illuminator (Analytik Jena US, Upland, CA, USA).

## Computational and statistical analysis

All statistical analyses in this paper were conducted using R (V 3.3) and computational analysis was performed using custom Perl and Python scripts. Fisher's r-to-z transformation was performed to test the difference between correlations using the cocor package in R[52]. The Fisher's exact test was done in R to test for enrichment of destabilized complexes with Chr16 or Chr8+15 subunits. Fisher's exact test was performed between the destabilized and stable protein complexes that contained subunits encoded by replaced chromosomes and protein complexes without subunits encoded by replaced chromosomes respectively. All the boxplots in this manuscript show the minimum, the maximum, the sample median, and the first and third quartiles of the data.

## Statistics and reproducibility

All the experiments presented here were repeated at least three times independently with similar results.

## Reporting summary

Further information on research design is available in the Nature Research Reporting Summary linked to this article.

## Data availability

The RNA-seq datasets generated in this study have been deposited in NCBI under the accession number BioProject PRJNA855266. The mass spectrometry proteomics data have been deposited in ProteomeXchange Consortium under the accession number PXD028358. The RNA-seq and the mass spectrometry proteomics data analyzed in this study are provided as Supplementary Data. Other data generated or analyzed in this study are provided in the Source Data file. All supplementary files and Source Data are provided with this article. Source data are provided with this paper.

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

## Acknowledgements

We thank members of the Leu lab for helpful discussions and comments on the manuscript. We thank Hsilin Cheng from the IMB Genomics Core for technical assistance. Mass spectrometry data were acquired at the Academia Sinica Common Mass Spectrometry Facilities for Proteomics and Protein Modification Analysis located at the Institute of Biological Chemistry, Academia Sinica, supported by Academia Sinica Core Facility and Innovative Instrument Project (AS-CFII-108-107). We thank the Sequencing Core of Academia Sinica for sequencing services, the Genomics, Bioinformatics, and FACS Cores of IMB for technical assistance, and John O'Brien for manuscript editing. We thank Gavin Sherlock for sharing pGS62 and pGS64 plasmids. H.Y.L. and K.B.S.S. were supported by Academia Sinica postdoctoral fellowships. K.B.S.S. was supported by the DBT-Ramalinagaswami fellowship (AU/SAS/DBT-RLS/20-21/04_KS_03.25) and Ahmedabad University start-up grant (AU/SUG/SAS/DBLS/2019-20/01). J.Y.L. was supported by Academia Sinica of Taiwan (grant no. AS-IA-110-L01 and AS-TP-107-ML06) and the Taiwan Ministry of Science and Technology (MOST 109-2326-B-001-015).

## Author contributions

J.Y.L. conceived the study. K.B.S.S., H.Y.L., and J.Y.L. designed analyses and interpreted results. K.B.S.S., H.Y.L., C.L., J.C.C., and Y.Y.C. performed the experiments. K.B.S.S., H.Y.L., and C.F.J.L. performed computational analyses. K.B.S.S., H.Y.L., and J.Y.L. wrote the paper. All authors read and approved the final manuscript.

## Competing interests

The authors declare no competing interests.
