## [Peer Review File · Nature Communications]

Proteotoxicity caused by perturbed protein complexes underlies hybrid incompatibility in yeastReviewers' Comments:

Reviewer #1:

Remarks to the Author:

The study examines proteotoxicity caused by chimeric protein complexes as a cause of hybrid incompatibility. This is a new angle on a long-standing question in speciation and hybrid biology and the results are likely to be of broad interest. I have no major concerns, although there are a few areas to clarify/improve.

1. Some of the replacement lines do not appear to be stressed (5+7, 12, 13, and 1 and 11 showing a weak pattern) based on their correlation with stress response expression. It would be worth discussing how the 5+7L having so little stress given that it has a large number of proteins in complexes and is among the more sensitive to GdA?

2. The correlation between GdA fitness (or HSP104 foci) and the number of complexes per chromosome doesn't control for the size of the chromosome, ie number of proteins. Saying that it is not directly related to chromosome length and pointing to one outlier (Chr 16) is not a strong argument. It is reasonable that proteotoxicity could be a function of the number of proteins (not complexes) on the substituted chromosome. This could occur if the chaperone/folding apparatus of *S. cerevisiae* doesn't work so well on some portion of *S. uvarum* proteins. I believe it is still worth reporting the correlation, but it would be important to also state whether this is significant once you control for the total number of proteins, e.g. test for a correlation with the ratio of proteins in a complex divided by total proteins on the chromosome.

3. The statements on Lines 354-356 are not clear in comparison to the Table S6. For example, 10 are destabilized in 16L, listed in Table S6a. 9/10 of these have at least one subunit on Chr 16. However, Table S6b lists complexes that are unstable after remove of Chr 16 subunits; 9 out of the 10 complexes were found to be significant. For Lines 354-356 do you mean: After removal of subunits on Chr 16, 9/10 complexes were still destabilized? I'm also unclear on what is being compared in Fisher's Exact test.

4. The results of these papers should be mentioned in the discussion (or introduction):

- Evidence for the Robustness of Protein Complexes to Inter-Species Hybridization
- Increased proteomic complexity in *Drosophila* hybrids during development

5. I understand the logic for Fig 4a, but the differences are very subtle between the boxplots. I believe it would be more informative to show relative abundance for proteins in the complexes as is shown in Fig 3c.

Reviewer #2:

Remarks to the Author:

This is an extremely broad and insightful manuscript by Sway et al. It builds on earlier observations from Jun-Yi Leu's lab on mapping specific incompatibility factors between different *Saccharomyces* species, which was itself a tour de force. However, the present manuscript is much broader and likely to have implications for hybrid dysfunction in general. The manuscript is exceedingly well written, easy to follow, and convincing.

There are several noteworthy findings:

- (1) Chromosome replacement lines (in which one chromosome has been replaced by the orthologous chromosome from a related species- a few sentences could describe this better in the Introduction) show a general signature of stress response transcriptionally relative to wildtype yeast.
- (2) This stress response is due to proteotoxic stress as shown by a quantitative assay of aggregate

formation that is exacerbated by Hsp90 inhibition.

(3) Hsp90 inhibition is much more detrimental to mitosis and meiosis in replacement lines. The comparison is important because Hsp90 has many client proteins including mitotic and meiotic complexes- a short sentence would be nice to present and then dispense with this possibility.

(4) levels of proteotoxic stress correlate with the number of protein complexes affected, shown by beautiful genetic, transcriptional and protein complex stability assays. Notably the stress is dominant i.e. can also be seen in diploid yeast with chromosome replacements. This makes this distinct from many of the similar proteotoxic stress findings made by Dunham and Amon labs previously- the authors state this clearly. This also translates to lower protein abundance in replacement lines and is not simply a function of their transcriptional consequences (full disclosure; the analysis to say that this is not purely transcriptional seems very robust to me but is beyond my expertise).

(5) Most strikingly, upregulating the protein degradation machinery alleviates significantly the fitness defects of replacement lines. This really makes this an airtight case.

There is a dizzying array of methods here and makes for a wonderful complete story. If I had one experiment to suggest (and this is not a recommendation the authors perform this)- they could take advantage of their hybrid diploids with chromosome replacement experiment and remove potential complex members that could lead to dysfunction. So if complex A is essential and comprised of components from both bayanus and cerevisiae chromosomes, one could not delete any bayanus component because the complex would be essential in haploid replacement lines. However, this deletion could be complemented in the hybrid diploid lines and may be less deleterious- thus deletion of complex subunits that lead to proteotoxicity would improve fitness. A version of this assay is already present in the paper which is why I certainly do not think it is needed for what is already a complete story.

This study is likely to have a fairly significant effect on the fields of proteostasis and speciation, much beyond yeast.

My primary expertise is in Drosophila speciation.

---Harmit S. Malik

1. Response to Reviewer 1 (Reviewer's comments in bold, responses in red):

Reviewer #1 (Remarks to the Author):

The study examines proteotoxicity caused by chimeric protein complexes as a cause of hybrid incompatibility. This is a new angle on a long-standing question in speciation and hybrid biology and the results are likely to be of broad interest. I have no major concerns, although there are a few areas to clarify/improve.

1. Some of the replacement lines do not appear to be stressed (5+7, 12, 13, and 1 and 11 showing a weak pattern) based on their correlation with stress response expression. It would be worth discussing how the 5+7L having so little stress given that it has a large number of proteins in complexes and is among the more sensitive to GdA?

Response: It is indeed intriguing as to why 5+7L cells with a large number of proteins demonstrate a mild correlation with stress but are sensitive to protein misfolding. Although the exact cause of this is unclear, one possibility is that in the absence of external stimulus, protein complexes in 5+7L cells are not severely perturbed as the average nonsynonymous substitution rate for proteins on Chromosomes 5 and 7 is relatively lower compared to proteins on Chromosomes 8, 15 and 16. Under congenial growth conditions, 5+7L cells can thus maintain homeostasis, which is impaired when cellular chaperone machinery is compromised or when the external environment escalates protein misfolding.

2. The correlation between GdA fitness (or HSP104 foci) and the number of complexes per chromosome doesn't control for the size of the chromosome, ie number of proteins. Saying that it is not directly related to chromosome length and pointing to one outlier (Chr 16) is not a strong argument. It is reasonable that proteotoxicity could be a function of the number of proteins (not complexes) on the substituted chromosome. This could occur if the chaperone/folding apparatus of *S. cerevisiae* doesn't work so well on some portion of *S. uvarum* proteins. I believe it is still worth reporting the correlation, but it would be important to also state whether this is significant once you control for the total number of proteins, e.g. test for a correlation with the ratio of proteins in a complex divided by total proteins on the chromosome.

Response: We thank the reviewer for the suggestion. We computed the correlation between the ratio of proteins in a complex divided by total proteins on the chromosome vs fitness defect under GdA and HSP104 foci, but the correlation was not significant ($p > 0.05$). We have added this information into the revised manuscript (page 10, line 294).

3. The statements on Lines 354-356 are not clear in comparison to the Table S6. For example, 10 are destabilized in 16L, listed in Table S6a. 9/10 of these have at least one subunit on Chr 16. However, Table S6b lists complexes that

are unstable after remove of Chr 16 subunits; 9 out of the 10 complexes were found to be significant. For Lines 354-356 do you mean: After removal of subunits on Chr 16, 9/10 complexes were still destabilized? I'm also unclear on what is being compared in Fisher's Exact test.

Response: The Fisher's exact test was done to test for enrichment of destabilized complexes with subunits encoded by replaced chromosomes. Fisher's exact test was performed between the destabilized and stable protein complexes that contained subunits encoded by Chr16 or Chr8+15 and protein complexes without Chr16 or Chr8+15 subunits respectively. In this analysis, we did not exclude the subunits encoded by replaced chromosomes (Supplementary Data 6a, originally named Table S6a). We also found that we made a mistake in our previous manuscript and the correct p-value is <0.05. We have added this information into the revised manuscript (page 13, line 367; page 27, line 799).

As the reviewer has correctly pointed out, Supplementary Data 6b is presented to emphasise that most of the destabilized protein complexes (8/10 Chr16 complexes and 19/20 8+15L complexes) were retained in 8+15L and 16L cells even after excluding the subunits encoded by replaced chromosomes. We have now clarified the difference between Supplementary Data 6a and 6b in the main text (page 12, line 337 and line 352; page 13, line 367).

4. The results of these papers should be mentioned in the discussion (or introduction): -Evidence for the Robustness of Protein Complexes to Inter-Species Hybridization -Increased proteomic complexity in Drosophila hybrids during development.

Response: We thank the reviewer for the suggestion. We have now cited these papers in page 16, line 477 and page 17, line 502.

5. I understand the logic for Fig 4a, but the differences are very subtle between the boxplots. I believe it would be more informative to show relative abundance for proteins in the complexes as is shown in Fig 3c.

Response: We thank the reviewer for this suggestion. We have now added a new figure (Figure S7a and page 25, line 739) that shows the complexes with significantly reduced subunit protein abundance.

2. Response to Reviewer 2 (Reviewer's comments in bold, responses in red):

Reviewer #2 (Remarks to the Author):

This is an extremely broad and insightful manuscript by Sway et al. It builds on earlier observations from Jun-Yi Leu's lab on mapping specific incompatibility factors between different Saccharomyces species, which was itself a tour de

force. However, the present manuscript is much broader and likely to have implications for hybrid dysfunction in general. The manuscript is exceedingly well written, easy to follow, and convincing.

There are several noteworthy findings:

(1) Chromosome replacement lines (in which one chromosome has been replaced by the orthologous chromosome from a related species- a few sentences could describe this better in the Introduction) show a general signature of stress response transcriptionally relative to wildtype yeast.

Response: We have added a short description of chromosome replacement lines in the introduction (page 4, line 103).

(2) This stress response is due to proteotoxic stress as shown by a quantitative assay of aggregate formation that is exacerbated by Hsp90 inhibition.

(3) Hsp90 inhibition is much more detrimental to mitosis and meiosis in replacement lines. The comparison is important because Hsp90 has many client proteins including mitotic and meiotic complexes- a short sentence would be nice to present and then dispense with this possibility.

Response: We have added short sentences emphasising the importance of studying mitosis and meiosis under Hsp90 inhibition in page 8, line 222

(4) levels of proteotoxic stress correlate with the number of protein complexes affected, shown by beautiful genetic, transcriptional and protein complex stability assays. Notably the stress is dominant i.e. can also be seen in diploid yeast with chromosome replacements. This makes this distinct from many of the similar proteotoxic stress findings made by Dunham and Amon labs previously- the authors state this clearly. This also translates to lower protein abundance in replacement lines and is not simply a function of their transcriptional consequences (full disclosure; the analysis to say that this is not purely transcriptional seems very robust to me but is beyond my expertise).

(5) Most strikingly, upregulating the protein degradation machinery alleviates significantly the fitness defects of replacement lines. This really makes this an airtight case.

There is a dizzying array of methods here and makes for a wonderful complete story. If I had one experiment to suggest (and this is not a recommendation the authors perform this)- they could take advantage of their hybrid diploids with chromosome replacement experiment and remove potential complex members

that could lead to dysfunction. So if complex A is essential and comprised of components from both bayanus and cerevisiae chromosomes, one could not delete any bayanus component because the complex would be essential in haploid replacement lines. However, this deletion could be complemented in the hybrid diploid lines and may be less deleterious- thus deletion of complex subunits that lead to proteotoxicity would improve fitness. A version of this assay is already present in the paper which is why I certainly do not think it is needed for what is already a complete story.

Response: We thank the reviewer for the positive response. As the reviewer has pointed out, we have already presented a version of the experiment suggested by the reviewer and hence would beg to refrain from performing the suggested experiment.

Reviewers' Comments:

Reviewer #3:

Remarks to the Author:

The study by Swamy et al. envisions elucidating the molecular mechanisms underlying genetic incompatibilities between species. To that end, the authors use previously established diploid chromosome replacement yeast cell lines to characterize the general consequences of mixed genomes. The authors conclude that reduced fitness in these cell lines is caused by proteostasis disruption by unstable chimeric protein complexes. First, the study provides sound evidence that the replacement cell lines have a proteostasis deficit: transcriptomic analysis revealed activation of stress response genes, microscopy experiments showed formation of heat-dependent Hsp104 foci diagnostic of protein aggregation, and replication analysis showed that cells were particularly sensitive to Hsp90 chaperone inhibition. These are straightforward, complementary methods that support this conclusion. In addition, it was demonstrated that these phenotypes can be alleviated by modulating proteostasis capacity, by means of genetic interventions on proteasome components.

The authors then attribute the proteostasis problem to the formation of unstable chimeric protein complexes. To show that chromosome replacements destabilize protein complexes, the authors compare the elution profile of soluble proteins in the control and replacement cell lines in gel filtration experiments, using mass spectrometry-based quantification. I find the methodology to be adequate and highly informative, as it monitors a broad range of proteins in an unbiased manner. The chosen mass spec methods are highly quantitative and follow standard, validated procedures in the field. The authors use published evidence to classify proteins into "complex" and "non-complex", and the corresponding classifications are reported in detail in the supplemental file, which I find quite commendable. By comparing the differences in the elution profile of chimeric and non-chimeric complexes, the authors nicely demonstrate that protein complexes containing chromosome-replaced subunits are destabilized, with no or very little effect on non-replaced complexes. It is also shown that the abundance of proteins from chimeric complexes decreases, which agrees with the widely accepted notion that surplus/unengaged proteins undergo rapid turnover. While it is very difficult (if possible at all) to establish a direct causal link between chimeric complex destabilization and proteostasis disruption, the authors show convincing correlations between the number of complexes on the replaced chromosomes and the observed proteostasis phenotypes. Therefore, I think the conclusion that protein complex destabilization plays a role in proteostasis failure and loss of fitness is justified. Overall, I think this is a solid and quite clever study and the conclusions are well supported and not overstated. As discussed by the authors, proteome balance is one of the most demanding tasks of the cell and a myriad of perturbations can affect it. My opinion is that a functional loss of multimeric chaperone/degradation systems is likely to play a major role. While it would be very interesting to follow-up on that, I think no further experiments are necessary for the current manuscript. I therefore recommend it for publication.

One correction: in line 312 it should be "Fig. 3" and not "Supplementary Fig. 3"

1. Response to Reviewer 3 (Reviewer's comments in bold, responses in red):

Reviewer #3 (Remarks to the Author):

The study by Swamy et al. envisions elucidating the molecular mechanisms underlying genetic incompatibilities between species. To that end, the authors use previously established diploid chromosome replacement yeast cell lines to characterize the general consequences of mixed genomes. The authors conclude that reduced fitness in these cell lines is caused by proteostasis disruption by unstable chimeric protein complexes. First, the study provides sound evidence that the replacement cell lines have a proteostasis deficit: transcriptomic analysis revealed activation of stress response genes, microscopy experiments showed formation of heat-dependent Hsp104 foci diagnostic of protein aggregation, and replication analysis showed that cells were particularly sensitive to Hsp90 chaperone inhibition. These are straightforward, complementary methods that support this conclusion. In addition, it was demonstrated that these phenotypes can be alleviated by modulating proteostasis capacity, by means of genetic interventions on proteasome components.

The authors then attribute the proteostasis problem to the formation of unstable chimeric protein complexes. To show that chromosome replacements destabilize protein complexes, the authors compare the elution profile of soluble proteins in the control and replacement cell lines in gel filtration experiments, using mass spectrometry-based quantification. I find the methodology to be adequate and highly informative, as it monitors a broad range of proteins in an unbiased manner. The chosen mass spec methods are highly quantitative and follow standard, validated procedures in the field. The authors use published evidence to classify proteins into "complex" and "non-complex", and the corresponding classifications are reported in detail in the supplemental file, which I find quite commendable. By comparing the differences in the elution profile of chimeric and non-chimeric complexes, the authors nicely demonstrate that protein complexes containing chromosome-replaced subunits are destabilized, with no or very little effect on non-replaced complexes. It is also shown that the abundance of proteins from chimeric complexes decreases, which agrees with the widely accepted notion that surplus/unengaged proteins undergo rapid turnover. While it is very difficult (if possible at all) to establish a direct causal link between chimeric complex destabilization and proteostasis disruption, the authors show convincing correlations between the number of complexes on the replaced chromosomes and the observed proteostasis phenotypes. Therefore, I think the conclusion that protein complex destabilization plays a role in proteostasis failure and loss of fitness is justified. Overall, I think this is a solid and quite clever study and the conclusions are well supported and not overstated. As discussed by the authors, proteome balance is one of the most demanding tasks of the cell and a myriad of perturbations can affect it. My opinion is that a functional loss of multimeric chaperone/degradation systems is likely to play a major role. While it would be very interesting to follow-up on that, I think no further

experiments are necessary for the current manuscript. I therefore recommend it for publication.

One correction: in line 312 it should be “Fig. 3” and not “Supplementary Fig. 3”

Response: We thank the reviewer for the positive comments and support. We have corrected the typo in the revised manuscript.